# Water Network Partitioning into District Metered Areas: A State-Of-The-Art Review

**Xuan Khoa Bui, Malvin S. Marlim and Doosun Kang \***

Department of Civil Engineering, College of Engineering, Kyung Hee University, Gyeonggi-do 17104, Korea; khoabx@khu.ac.kr (X.K.B.); malvinmarlim@hotmail.com (M.S.M.)

**\*** Correspondence: doosunkang@khu.ac.kr; Tel.: +82-31-201-2513

**Abstract:** A water distribution network (WDN) is an indispensable element of civil infrastructure that provides fresh water for domestic use, industrial development, and fire-fighting. However, in a large and complex network, operation and management (O&M) can be challenging. As a technical initiative to improve O&M efficiency, the paradigm of "divide and conquer" can divide an original WDN into multiple subnetworks. Each subnetwork is controlled by boundary pipes installed with gate valves or flow meters that control the water volume entering and leaving what are known as district metered areas (DMAs). Many approaches to creating DMAs are formulated as two-phase procedures, clustering and sectorizing, and are called water network partitioning (WNP) in general. To assess the benefits and drawbacks of DMAs in a WDN, we provide a comprehensive review of various state-of-the-art approaches, which can be broadly classified as: (1) Clustering algorithms, which focus on defining the optimal configuration of DMAs; and (2) sectorization procedures, which physically decompose the network by selecting pipes for installing flow meters or gate valves. We also provide an overview of emerging problems that need to be studied.

**Keywords:** clustering; district metered area; network sectorization; water distribution network; water network partitioning

---

## 1. Introduction

A water distribution network (WDN) supplies drinking water by maintaining pressures and flow rates. As most of a WDN's components are buried and comprise thousands to tens of thousands of elements, operation and management (O&M) can be complex [1]. Increasing urbanization means WDNs are constantly being upgraded and expanded. In large cities with aging networks, O&M is becoming more challenging than ever before.

A critical O&M objective for utilities working on WDNs is improving the efficiency and efficacy of the supply for a specified water demand at the lowest cost. In particular, efficacy requires reducing water leakage and nonrevenue water, controlling uniform pressure, and ensuring sufficient pressure. Leakage control is the most effective way to reduce water prices. The quantity of leakage is related to system pressure, and reducing pressure reduces leakage. Utilities can apply a "divide and conquer" paradigm to this challenge by dividing the original complex network into independently controlled subnetworks called district metered areas (DMAs).

Most researchers agree that partitioning a network into DMAs offers many benefits [2–6]. These actions may include but are not limited to: (i) Substantially reduce nonrevenue water by active leakage control; (ii) simplify pressure management by setting off pressure reducing valves (PRVs); (iii) rapidly identify burst pipes; (iv) district isolation in order to protect the rest of network from accidental or malicious contamination events; and (v) potential creation of independent DMAs which exclusively supplied from its own water sources for better control of water quality (e.g., there is no mixing of

water from different sources). Moreover, for intermittent WDNs, where water is only supplied during a certain time of a day, DMAs are transition processes that allow evolving intermittent WDNs to continuous systems by enabling equitable water supply in each DMA [7]. Ciaponi et al. [8] recently revealed the benefits of WNP not only for WDNs monitoring from contamination events but also for the effectiveness of optimal sensor placement.

Despite these advantages, they come with trade-offs, such as the reduced redundancy in network connectivity and the demotion of system pressure, which results in lower network preparedness for emergencies such as fire-fighting, water suspension due to burst pipes. An additional concern is water quality deterioration (i.e., water age growth) due to the reduction of available flow paths [9,10].

Because of the benefits brought about by DMAs, many utilities consider them an effective way to achieve O&M objectives [2]. However, dividing an original network into suitable DMAs can be challenging because of the intrinsic complexity of the WDN. In the past, before mathematical methods were applied to DMA configuration, utilities designed DMAs according to administrative boundaries (districts), main roads, the number of inhabitants, the economic level of leakage, or reservoir (tank) locations [4,5], which did not account for global perspectives. However, with the advent of mathematical models, hydraulic solvers can simplify the process and provide various approaches to optimizing the creation of DMAs while considering operational constraints and objectives.

Today, water network partitioning (WNP) is used to divide networks into DMAs. WNP is a heuristic process controlled by two phases: clustering and sectorization. The clustering phase is the preformation of DMAs based on network connectivity and topology. It is implemented through various algorithms such as graph theory, community structure, modularity-based algorithms, and spectral algorithms [2,3] to form feasible DMAs and minimize the number of connections to each other. Sectorization is an optimization process to locate flow meters and gate valves to maintain as high as possible network performances while minimizing the economic costs [6].

In recent years, WNP has been explored in various studies. In the 18th Water Distribution System Analysis Conference held in Colombia in July 2016, the "Battle of Water Networks" competition focused on creating DMAs. The main objective was to optimize the design and operation of a system's main components by determining new DMAs for an existing WDN in Colombia, the E-Town network, by taking into account costs, pressure uniformity, and water quality [3]. It was an opportunity for researchers and practitioners around the world to solve a challenging problem in a full-scale WDN. In addition, several methods have been proposed over the last decade for dividing a network into DMAs. Various benchmark networks have also been developed to test the state-of-the-art methods.

A literature review identified more than 100 published studies that focused applying various methodologies to WNP. After reviewing the main discussions and approaches in each paper, we selected 95 papers to study in-depth and 27 related articles; they are all cited here. We found that the methodologies proposed to date still have certain limitations in real-life applications. This paper provides a comprehensive review of WDN management using DMAs to help water utilities improve efficiency in O&M as well as support decision- and strategy-making processes. With this goal in mind, we first reviewed the rules of DMA and analyzed the merits and demerits of the O&M of DMAs. Second, we classified the methodologies proposed in recent studies of WNP processes into two major categories, clustering algorithms and sectorization processes, and analyzed the advantages and disadvantages of each method. Then, we highlighted the main indicators developed to quantify the segmentation performances. Finally, we discussed some limitations of the approaches.

This paper is organized as follows. Section 2 presents the principle of DMA and its function in the O&M of WDNs. Section 3 mainly reviews the clustering phase for the methods developed so far, and Section 4 describes the sectorization procedure and discusses the currently applied algorithms. Various indicators of DMA performance are described in Section 5. Finally, Section 6 draws conclusions and discusses possible improvements in WNP approaches.

## 2. Principles of District Metered Areas

The concept of DMA management was introduced by the United Kingdom water industry in the early 1980s [2,5,11] (Figure 1). At that time, DMA was an area of a distribution system that was specifically defined by the closure of valves and measurement of the quantities of water entering and leaving the area. The first goal of DMA is early detection and management of water leakage in a WDN [11]. Specifically, the measurement of night flow is analyzed to determine the level of leakage within each DMA and locate the most beneficial places for leakage probes [4,10].

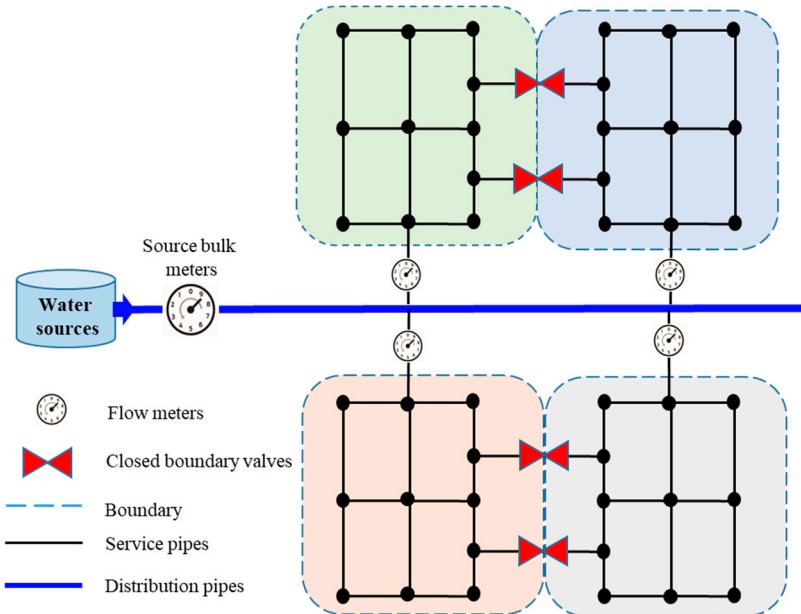

**Figure 1.** A schematic diagram of district metered areas.

Water leakage is a major concern for water utilities [12]. Leakage rate varies widely depending on the country, region, and age of the system. It is reported to be as low as 3 to 7% in a well-maintained system in the Netherlands, ranging from 10 to 30% in the United States and the United Kingdom, however as high as 70% in some undeveloped countries [13]. Water loss in WDNs can be classified as real loss and apparent loss in nature [14]. The real loss occurs from burst pipes or background leaks due to continual seepage of water from network properties, such as pipe and valve fittings, or to corrosion-induced perforation of pipes. The apparent loss includes the unauthorized consumption, a product of meter-reading errors, water theft, and accounting errors. To estimate the leakage in a DMA, the operator must monitor net minimum night flows in the system (when most consumers are inactive) and compare it with legitimate night flows to assess the rate of real losses.

One of the major factors influencing leakage is the pressure in the water network [12]. To reduce water losses, many utilities have changed from a passive approach (i.e., detection and repair) to proactive approaches (i.e., heuristic processes and pressure-leakage relationships as developed by Allan Lambert [15]) that indicate that the leakage rate of flow may increase or decrease with changing pressure levels. The DMA concept was introduced to help proactively manage the number of invisible water losses and detect the locations of failures based on the hydraulic characteristics of the WDN.

Researchers agree that dividing a network into DMAs is useful [4,16]. Most research assumes that the benefits of DMAs are greater than the drawbacks [6,17–21]. WDN management through DMAs has proven highly successful for leakage reduction, reportedly controlling up to 85% of national water leakage in the UK [11,22]. Gomes et al. [23] showed that dividing a network into DMAs allows for stable pressure management, which increases asset lifespans. Reduced pressure lowers the frequency of potential pipe breaks, which consequently reduces real water losses. Gomes et al. also proposed a method based on the minimum night-flow relationship with pressure to evaluate the

benefits of pressure management using DMAs by predicting water losses before and after pressure drops, estimating the reduction in energy consumption through billed water as well as the estimated direct benefit of the pressure reduction process with DMAs [23]. Huang et al. [24] reported that DMAs allow for rapid detection of burst pipes by studying the uniformity of daily water demand. They applied a supervised learning algorithm to improve the positive effect of burst-event detection in real-time operations. Savic and Ferrari [20,25] and Lifshitz and Ostfeld [26] have also illustrated the effectiveness of implementing DMAs in WDNs with respect to reducing the frequency of pipe breaks. To quantify the benefits of reducing burst frequency, Lambert et al. [15] proved that the percentage of burst-frequency reduction relies on the proportion of pressure reduction obtained after setting up the DMAs. Their study also revealed that controlling pressure not only reduces burst frequencies, but also reduces leakage flow rates, extending the life of residual devices and reducing costs for both the water utility and the customer.

Ferrari and Savic [25] showed that, depending on the specific alternative DMA layout used, burst frequency can be reduced by approximately 53% to 60%. They also found that leakage reduction ranged between approximately 26% and 59% after DMA set-up. Furthermore, as the closure of valves completely isolates DMAs, it is possible to reduce the risk of chemical attacks or accidental events throughout a network [6]. Isolating DMAs is also useful in component maintenance, replacement, and repairs because closing boundary valves disconnects districts from other areas. Lifshitz and Ostfeld [26] demonstrated that combining DMAs with PRVs creates a "knowledge and action" approach to detecting and managing water leaks. PRVs reduce excess pressure and consequently reduce potential water leaks without prior information on the positions of the leaks. Meanwhile, DMAs enable the identification of possible locations of leaks and their potential amounts. A combination of the DMAs and PRVs will complement each other to provide a better solution for leak management.

The main drawbacks of DMAs are deterioration of water quality compared with that of the original network and the loss of system resilience against abnormal events. Water age is regarded as a surrogate for simulations for evaluating the reduction of water quality [27]. Grayman et al. [10], Diao et al. [19], and Di Nardo et al. [28] found that there was no significant change in the overall water age metric before and after dividing a large-scale, looped WDN into DMAs. This is consistent with previous studies by UKWIP [29] and WRc [30], which investigated the impact of WNP on water quality management. Armand et al. [31] utilized surrogate hydraulic variables to evaluate the impact of WNP on water quality and the likelihood of discoloration incidents. They reported that DMAs can compromise overall water quality by increasing the average water age for a set of nodes with dead-end-like hydraulic behavior. This also increased the likelihood of sedimentation in pipes due to low flow velocity. However, water quality is reportedly not a critical criterion when designing DMAs and water age is not a binding constraint [3,28]. Javier et al. [32] and Salomons et al. [33], who conducted water balance analyses in a WDN, pointed out that the water volume stored in the network was nearly half of the daily water consumption. It was therefore reasonable to assume that water would be replaced twice a day in the network, which is a good indicator of water quality. By running a hydraulic model to compare the network before and after DMA installation, no significant variations of water age were seen throughout the whole network.

One of the other weaknesses when dividing networks into DMAs is the reduction in a system's redundancy [19,28] due to reduced availability of flow paths to connect supply sources and demand nodes. The insertion of multiple gate valves and flow meters to isolate a DMA leads to increased head loss due to increased friction [34]. This change can reduce system redundancy in terms of available pressure throughout the network. Typically, several emergent cases, such as fire-flow supply and water suspension due to a burst pipe would be issued in system operations. Table 1 summarizes the main advantages and disadvantages of installing DMAs in WDNs.

**Table 1.** Main advantages and disadvantages of district metered areas (DMAs).

| Advantages | Disadvantages |
|---|---|
| Improved burst detection and leakage identification | Reduced resilience to failures |
| Advantaged subarea management and reduced NRW | Reduced operational flexibility |
| Improved subarea pressure control | Potential negative impact on water quality |
| Improved protection against contamination | Security issues in peripheral areas and emergency cases |
| Reduced maintenance and repair costs | High initial investment cost |
| Characterized demand curve, especially at night | Reduced hydraulic redundancy |

Several criteria should be considered when designing DMAs [11], such as

1. Maximum percentage of leakage allowed by the water utility;
2. Topography and number of properties per DMA;
3. Characteristics and topological taxonomy of WDNs;
4. Variations in nodal elevation, water demand, and pressure;
5. The number of flow meters and gate valves; and
6. Water quality considerations.

Depending on the existing network situation and leakage rates, each utility will have its own criteria to set up economically efficient levels of leakage for each DMA. Once the level of leakage has been determined, the utility can select the type of policy best suited for controlling leakage in the future, the size and number of DMAs, and the staff required for the required policy. Dividing a network into small DMAs will identify bursts quickly, maintain total leakage at a lower level, and reduce the time required to identify device failures. However, this also leads to increased investment and operational costs in terms of new flow meters and valves [11]. The international water association (IWA), as corroborated by previous studies, reports that DMA size is expressed by the number of properties (user flow meters) and varies between 500 and 5000 properties in urban areas [24]. Individual DMA size can vary depending on local factors and system characteristics. While a DMA with fewer than 500 properties requires much more initial investment and incurs a higher maintenance cost, a large DMA will face difficulty in discriminating small bursts and will suffer increased leakage location times [4,5,11].

From a topological connectivity point of view, a set of complex network metrics was proposed by Giudicianni et al. [35] to analyze the relationship between the metrics values and the topological structures of WDNs. To optimize the number of DMAs in the network, the eigengap heuristic was used to maximize the jump in spectrum of the Laplacian matrix. The study revealed that correlation between the number of DMAs and network size approximatively follows a power law. Hence, the optimal number of DMAs does not grow significantly with the network size. Such a relationship hints that, from a connectivity point of view, the increase of WDN size has more effects on the size of the DMA rather than the number of DMAs.

The number of water sources supplying each DMA also needs to be considered in the design process, as each source must be fitted with a flow meter. Depending on the network type (branched or looped), a DMA may be supplied by single or multiple sources and delivered consecutively or in parallel to adjacent DMAs. As suggested by Di Nardo et al. [2], a technical and economic rule is to minimize the number of installed flow meters and have a single flow meter for each DMA to simplify the calculation of the synchronous water balance. To isolate a DMA from adjacent DMAs, gate valves are installed in boundary pipes. However, installing gate valves may create more dead-ends and reduce the pathways of water to the nodes, which may lead to deteriorating water quality [11]. Therefore, optimizing the number and location of flow meters and valves while decomposing the original network into DMAs is necessary to minimize costs and optimize operational benefits.

Determining and optimizing the number of DMAs is essential. However, defining the configuration of DMAs is a demanding task because many different aspects of WDN performance must be considered [20]. This is usually approached as a multi-objective optimization problem. Traditionally, DMA design has been based on empirical data combined with trial-and-error methods. Recently, the concept and approach for WNP have been explored in the literature. Several smarter and more efficient approaches have been proposed to create optimal DMA layouts. Although the algorithms applied in each study are different, the WNP process commonly consists of two phases, clustering and sectorization [36,37].

## 3. Clustering to Create Feasible DMAs

Figure 2 summarizes the general procedures for WNP. The clustering phase is the initial process that designs the shape and dimensions of DMAs based on the network topology. The goal is to determine the optimal number of DMAs to balance the number of nodes in each cluster and to minimize the number of boundary pipes (i.e., pipe cuts where gate valves or flow meters will be installed). The algorithms applied include graph theory such as depth-first search (DFS) and breadth-first search (BFS) [6,9,38,39], community structure [19,34,37,40], modularity-based procedures [41–44], multilevel partitioning [17,37,45,46], spectral approaches [47–49], and multi-agent approaches [50–52]. This paper focuses on explaining six major algorithms and how they are handled in clustering WDNs to automatically create DMA configurations.

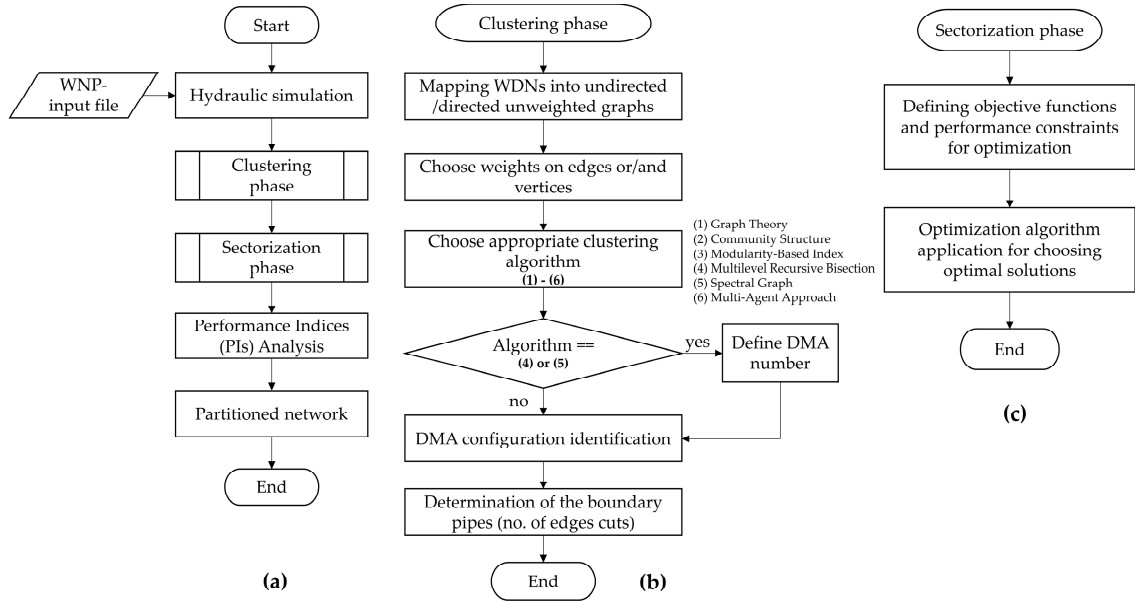

**Figure 2.** Steps of water network partitioning: (**a**) Overall main procedures, (**b**) steps for clustering, and (**c**) steps for sectorization.

### 3.1. Graph Theory

Most of the existing clustering algorithms developed for WNP relate to graph theory. To gain a deeper understanding of clustering algorithms, it is necessary to generalize some of the topological characteristics and properties of a WDN. Readers unfamiliar with graph theory should refer to previous studies [53,54].

WDN is a social infrastructure that allows water to flow along the pipes and communicate between nodes in the network. The topology structure of WDNs is mapped onto an undirected or directed graph and characterized by a pair of sets $G = (V, E)$, where $V$ is the vertex set representing junctions, reservoirs, and tanks and $n = |V|$ is the total number of vertices. $E$ is the set of edges in response to pipes, valves, pumps and $m = |E|$ is the total number of edges. An undirected graph with edges is an

unordered pair $\{v_1, v_2\}$, while a directed graph with edges is an ordered pair and the vertices $v_1, v_2$ are called the endpoints of the edge.

A given network graph, and a WDN in particular, can be converted by an adjacent matrix **A**, which is an $n \times n$ matrix, where $A_{ij}$ is the $(i, j)$ element equal to 1 if $v_i$ is adjacent with $v_j$, otherwise, $A_{ij} = 0$. A weighted graph can be represented mathematically by an adjacency matrix that has a certain weight $W_{ij}$ assigned for each pair of vertices $(i, j)$. The weights are usually non-negative, real numbers, and they must satisfy $W_{ij} = W_{ji} \geq 0$, if $i$ and $j$ are connected. Otherwise, $W_{ij} = W_{ji} = 0$. The nodal degree, $k_i$, is the number of edges attached to a vertex $i$. The degree of node $i$ is defined as $k_i = \sum_{j=1}^{n} A_{ij}$ for the adjacency matrix **A**, and $k_i = \sum_{j=1}^{n} W_{ij}$ for the weighted adjacency matrix **W**. From a topological point of view and complex network theory, Giudicianni et al. [35] treated the WDN as a graph by using several complex network metrics to characterize the topology of typical WDNs. It was a preliminary process for better understanding the network itself, and provided the classical approach for partitioning or/and designing the WDNs.

One of the graph theory algorithms applied to network clustering is DFS, which, as proposed by Tarjan [55], allows for the exploration of the connectivity of a graph by traversing a node in the network. The DFS algorithm is a recursive approach based on backtracking. It starts by picking a root node in the network and then searches for nodes as far as possible along each path (in-depth dimension) until there are no more adjacent nodes in the current path to traverse after backtracking to the next path. In contrast, the BFS algorithm proposed by Pohl [56] starts at a root node and traverses the graph broad-wise by first moving horizontally and exploring all the nodes of the current path and then moving to the next path. Figure 3 shows how a DFS and a BFS work.

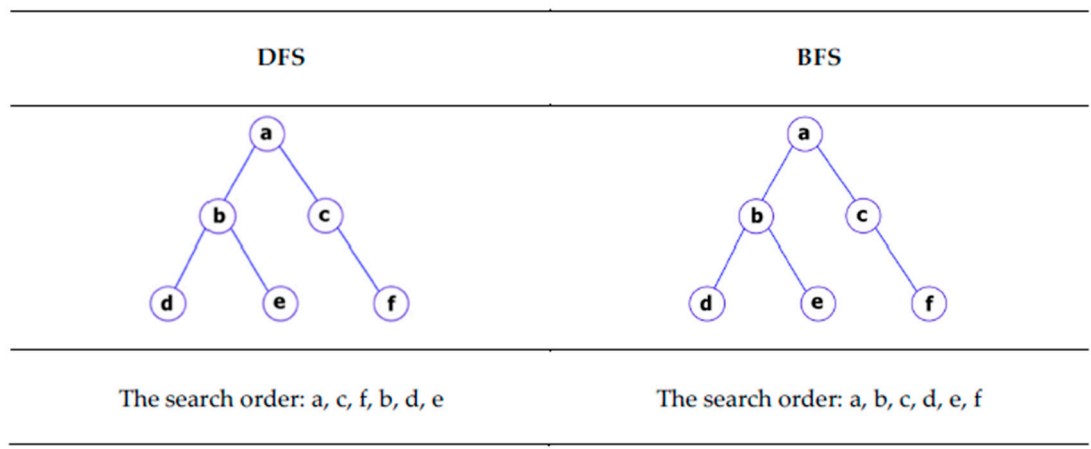

**Figure 3.** Diagram of depth-first search (DFS) and breadth-first search (BFS).

Tzatchkov et al. [38] applied the DFS and BFS to a WNP project in Mexico. DFS was used to segment a whole network into independent sectors by identifying nodes belonging to each sector (i.e., each sector is supplied exclusively from its own water sources, and it is not connected to other sectors in the network), and BFS was used to exam the set of disconnected nodes from any water sources, thus obtaining the size and configuration of independent sectors in the WDN. More specifically, Perelman and Ostfeld [39] and Lifshitz and Ostfeld [1] proposed a procedure for topology clustering based on the DFS algorithm to identify strongly connected clusters that had at least one path in both directions between them, while the opposing BFS algorithm was used to classify weakly connected clusters that had only one directed path between a set of nodes (i.e., from node $u$ to node $v$, but not from node $v$ to node $u$). The results were utilized for various purposes, such as contaminant prediction from a source and spread of infection in a WDN [1]. Di Nardo et al. [6] proposed a method for optimizing water network sectorization based on graph theory. DFS was used to find the independent sector combined with a hierarchical approach developed by Di Battista et al. [57] to draw hierarchical levels of a tree

graph corresponding to each source, creating isolated DMAs, each of which was supplied by its own source and was disconnected from the rest of a network through gate valves.

Campbell et al. [58,59] proposed a more advanced orderly combination of a series of graphs to generate a flexible method for defining feasible DMA layouts. They proposed dividing a network into two components, a trunk network and a distribution network. To determine the scope of the trunk network, the shortest path and the BFS concept were implemented. Once the trunk network was determined, it was detached from the network, while the community detection algorithm was adopted for the rest of network (distribution network) to define the best structural communities in the distribution network, which is the configuration of sectors. The innovation of this study was that the trunk pipes acted as entrances to each DMA and were not considered candidates for sectorization, ensuring the reliability of the WDN. In a similar methodology proposed by Alvisi and Franchini [60], BFS defined the location of possible nodes to form an assigned number of DMAs and then the shortest path distance from each source to the nodes was simultaneously estimated to determine the set of boundary pipes for each DMA. In the case of a WDN with numerous water sources, Scarpa et al. [9] successfully applied a BFS algorithm to identify elementary DMAs in which each one was supplied only by its own source.

Gomes et al. [61] proposed a systematic way to divide a WDN into suitable DMAs based on the Floyd–Warshall algorithm [62] and user-defined criteria (e.g., pipe length and number of users in each DMA). This method facilitated the creation of appropriate DMAs by finding the shortest distance from source to nodes depending on the network flow direction at peak flow conditions. Compared with BFS, this algorithm provided superior results, as it considered the shortest path of sources to every other node in the network and identified the best path. The algorithm was repeated until the target number of DMAs and user-defined constraints were met. Further adjustment could be carried out by combining adjacent DMAs to minimize the number of boundary pipes as long as the user-defined criteria are fulfilled.

*3.2. Community Structure Algorithm*

The community structure detection algorithm is a bottom-up hierarchical approach based on graph theory and proposed by Newman and Girvan [63] and Clauset et al. [64]. It uses greedy optimization of a quantity known as modularity ($Q$), which is defined in Equation (1). They used the quality measure of network density to define the clusters, assuming that the density of a network division was effective if there were many edges within communities (intraclusters) and only a few between them (interclusters). Modularity index is a network property used as an indicator to quantify the quality of graph division in the community. The clustering method is based on maximizing the modularity index. Higher values of that metric are related to a community structure of the network, which is significant if $Q \geq 0.3$ [63,64]:

$$Q = \frac{1}{2m} \sum_{ij} \left[ A_{ij} - \frac{k_i k_j}{2m} \right] \delta\left(C_i, C_j\right) \tag{1}$$

where $\delta\left(C_i, C_j\right)$ is the Kronecker delta coefficient, and $\delta\left(C_i, C_j\right) = 1$ if vertices $i$ and $j$ are the same community; otherwise $\delta\left(C_i, C_j\right) = 0$.

If we assume that the fraction of pipes that have both start and end nodes belonging to the same community is $e_{ii}$, and $a_i$ is the portion of pipes with at least one end node in the community $i$, then the modularity can be formulated as:

$$Q = \sum_c e_{ii} - a_i^2. \tag{2}$$

The change in the two communities $i$ and $j$ to increase modularity can be computed by [63]:

$$\Delta Q = 2\left(e_{ij} - a_i a_j\right). \tag{3}$$

The community structure algorithm is implemented following the steps listed in Figure 4.

Step 1: Concept of WDN as a graph $G = (V, E)$.
Step 2: Define adjacency matrix **A** or weighted undirected adjacency matrix **W**.
Step 3: Start with each node as a community.
Step 4: Compute the change in modularity $\Delta Q$ resulting from merging each pair of communities.
Step 5: Merge the pair of communities with the highest value in modularity change $\Delta Q$.
Step 6: Repeat steps 4 and 5 until one community remains.

**Figure 4.** Main steps for community structure algorithm clustering.

Diao et al. [19] first applied a community structure algorithm to detect clusters in a WDN. Their study used a community structure to automatically create boundaries for DMAs. WDN was mapped onto an undirected graph and community detection was implemented to maximize the modularity matrix and find the hierarchical community structure that represented the DMAs of the WDN. In the study, the authors determined the size constraint to be 300–5000 properties [10] for each community by applying a heuristic approach, known as oriented dendrogram cutting.

Instead of identifying network communities by maximizing the modularity index, Campbell et al. [34] proposed a procedure based on the idea that feedlines (i.e., a trunk network) should not be included in sectorization schemes. This was identified by means of determining the "betweenness" of edges, the flow, and the diameter analysis. The betweenness algorithm is a branch of graph theory that defines the edge (i.e., pipe) that connects to many pairs of vertices (i.e., nodes) [63]. A random-walk betweenness [19] can detect community segmentation with the highest modularity and a dendrogram can set the size constraint for each community.

Similarly, Ciaponi et al. [40] offered a different approach that combined convincing practical criteria when designing DMA as proposed by Morrison et al. [4]. Accordingly, automated identification of DMAs was performed by identifying the prevalent transport service (main transmission pipes) in WDNs and then each DMA, which was determined by the remaining distribution service pipes, was directly connected with the main transmission pipes. The procedure decomposed subsystems exceeding the threshold DMA size constraint owing to a modularity-based optimization algorithm. The two approaches brought the boundaries of identification of DMAs closer to reality and supported feasible alternative solutions to make more convincing decisions.

### 3.3. Modularity-Based Algorithm

The community structure algorithm uses a modularity index as a metric for the optimal design of DMAs. However, the modularity index may not be representative for the WDN because it is strongly affected by hydraulic properties (e.g., elevation, node demand, pipe diameter). Adopting the classic formulation of a modularity index without considering the physical and hydraulic constraints would therefore be artificial and misleading. Inspired by this approach, Giustolisi and Ridolfi [41] proposed a modularity-based method for WDN segmentation that accounts for hydraulic network properties to define WDN-oriented modularity. First, to formulate the modularity index for WDNs, the proposed method focused on conceptual segmenting of the network close to the ending nodes by using a topological incidence matrix and the number of pipes separating communities. This was done to minimize the number of required pipe cuts. Despite being tailored for a WDN, WDN-oriented modularity had an inherent limitation left over from the classic community detection algorithm. Fortunato and Barthelemy [65] stated that the modularity index proposed by Giustolisi and Ridolfi may fail to detect small communities if the community's total edge number is smaller than $\sqrt{2m}$, where $m$ is the total number of edges in the network. To overcome such failures, Giustolisi and Ridolfi [66] proposed that an infrastructure modularity index can improve the negative effect of the inconsistency of modularity optimization. A new index is released through maximization of the classic modularity index in the framework of the two-objective optimization, modifying the framework

to overcome the resolution limit. Laucelli et al. [67] took a further step by developing a flexible procedure for DMA planning based on Giustolisi and Ridolfi's achievement with a conceptual cut for segmentation. A two-step strategy was adopted for optimal sectorization design by maximizing the WDN-oriented modularity index versus minimizing the number of conceptual cuts, where the location of pipe cutting minimizes the number of devices to be installed. To determine the location of flow meters and gate valves, DMA design was optimized based on each conceptual cut and returned an optimal solution for each one, accounting for hydraulic behavior change in the network with respect to maximizing the reduction of background leakage in each DMA. Using the WDN-oriented modularity index, Simone et al. [68] developed a sampling-oriented modularity index to perform optimal spatial distribution and assess the optimal number of pressure meters needed in a network (i.e., sampling design) using a multi-objective optimization method to minimize pressure-meter cost versus sampling-oriented modularity.

As mentioned in Section 2, DMAs are designed to detect and actively manage leaks. To that end, pressure management is a fundamental and important factor affecting leak management. Zhang et al. [42] developed a hybrid procedure by combining node pressure with modularity-based community detection to segment a network into similar DMAs from a pressure aspect. However, to improve the resolution limits of classical modularity, they used a random-walk theory similar to that of Campbell et al. [34]. The random-walk theory allows for precise identification of communities with greater or smaller differences in size and the automatic creation of a multiscale community [42]. To illustrate the superiority of this method over previous methods, the results proposed by Diao et al. [19] were compared. They demonstrated that different partition schemes result at a variety of random-walk time periods because the variances of node pressure are integrated into the community. In Diao et al. [19], variance was made immutable using a top-down search. Additionally, in the aspect of boundary pipes proposed by the two respective methods, Zhang et al. [42] showed that the traditional modularity-based community detection introduced by Diao et al. had more boundary pipes.

Most recently, Perelman et al. [37] combined three branches of graph theory to evaluate the performance of each method. Global clustering, community structure, and graph partition were applied to two WDNs in Singapore. Global clustering is a bottom-up algorithm for grouping points concerning a measure of similarity defined for each pair of points. Community algorithms detect the community structure in the network focused on the concept of edge betweenness. Graph partitioning divides the graph into a predefined number of groups such that the number of edges crossing between the groups is minimal [69]. The authors showed that the methods were compatible and applicable to large networks, but the performance of each method was completely different and depended on the number of clusters and the parameters selected for evaluation. They proposed multi-criteria metrics based on visual and quantitative performance measures. Accordingly, a better approach would be to minimize four metrics, such as (a) worst cut size, (b) total cut size, (c) cluster size, and (d) running time, and maximize the metric in regard to (e) recurrence of inter-cluster edges [37]. The results demonstrate that graph-partitioning generally outperforms clustering and the community structure methods in terms of (a), (b), and (d), which implies that the number of flow meters needed to monitor the flow will be minimized. On the contrary, the global clustering method indicated a good expectation in terms of (e), while in terms of (c), the three methods showed similar results. Therefore, community structure and the graph partitioning methods were more flexible and outperformed global clustering under particular budget constraints.

Similarly, Di Nardo et al. [70] conducted a comprehensive analysis of two popular clustering algorithms, such as the graph partitioning based on multilevel recursive bisection (MLRB) and the spectral clustering based on the normalized cut algorithm. Applications to a real-life WDN in South Italy revealed that the graph partitioning outperformed the spectral clustering in balancing the number of nodes in each DMA. On the contrary, the spectral algorithm showed better performance than the graph partitioning to minimize the number of edges cuts, thus it was more efficient in both hydraulic and economic aspects. A similar study conducted by Liu et al. [71] explored the performance of three

partitioning methods, including fast greedy [64], random walk [63], and multilevel recursive bisection (MLRB) [72] using a spectrum of topology-based indicators.

As mentioned earlier, WDNs exhibit dynamic hydraulic behavior changes in the spatial and temporal mode that are completely different compared to others. Most of the partitioning algorithms lack exhaustive analyses of the similarity of the hydraulic and physical aspects in DMAs, such as the number of nodes and balance in terms of water demand and pressure. It is therefore not sufficient to offer a universal O&M solution to a utility. Awareness will make DMA segmentation more reliable when physical properties and hydraulic behavior are considered in network partitioning. Realizing the limitations of Diao et al. [19] and Ciaponi et al. [40], Creaco et al. [73] incorporated engineering aspects (i.e., demand supplied along the pipe and pipe length) into WNP processes. However, unlike Giustolisi and Ridolfi [41,66], they focused on applying heuristic procedures to improve the original fast greedy partitioning algorithm to maximize the modularity index developed by Clauset et al. [64]. Two heuristic optimization techniques were developed and applied to the formulation of modularity to perform different merging combinations. In the first technique, randomness was added to the DMA merging process, which allows for the acquisition of numerous WDN-partitioning probabilistic solutions while generating a higher modularity increment during the merging steps and a lower number of boundary pipes compared with the traditional deterministic approach. The second technique illustrated the trade-offs between various engineering aspects by embedding the former technique inside a multi-objective genetic algorithm optimization [74].

Evaluation of DMAs scenarios after sectorization must also guarantee that hydraulic indicators are at an acceptable or higher threshold compared with the original network. Because different criteria lead to various DMA layouts, Brentan et al. [43,44] proposed a method that considers the relationship between many technical criteria, such as demand and pipe length, to create different DMA scenarios. The social community detection algorithm was used to define DMAs. To assess the performance of DMA generation, a comprehensive analysis was proposed that considered performance indicators such as resilience index, demand similarity, pressure uniformity, water age, cost, and energy consumption, hopefully provides decision-makers with an optimal DMA configuration.

### *3.4. Multilevel Graph Partitioning*

Multilevel partitioning [72] is a fundamental approach based on an analogy of graph theory and graph-partitioning principles that uses parallel computing to allocate workloads among processors to minimize communication and equally distribute the computational burden among them. Based on that approach, the objective is to create subzones by equally distributing loads, such as DMA size, pipe length, water demand, and flow [45]. Recently, much effort has been devoted to developing techniques and heuristic procedures for optimal segmentation of a water network into isolated DMAs by balancing pipe length, nodal demand, and flow within each DMA [16,37,45,46].

Sempewo et al. [45] presented an automated prototype tool for the analysis of network spatiality to create analogy subzones based on balancing pipe length and demand at each zone using distributed computing called multilevel recursive bisection (MLRB) for monitoring and controlling leakage in the WDN. The core purpose of the MLRB algorithm was to design a highly effective method to deal with parallel *k*-way partitioning of a graph in computer science. The successor application of MLRB was mentioned by Di Nardo et al. [17,18,75]. They proposed a procedure adapting the traditional phase of the MLRB to create an automated tool for smart water network partitioning (SWANP). In the MLRB algorithm, three phases illustrating the computation of *k*-way partitioning in the graph are: coarsening, partitioning, and uncoarsening with refinement [72]. The coarsening phase simplifies the original graph by collapsing adjacent vertices in terms of maximally matching a graph with different techniques. The next phase is partitioning. First, the network is subdivided into a two-way partition. Each subgraph is then divided into bisections to obtain *k*-way partitioning. The boundary pipes that have the start and end nodes in different subgraphs must be minimized for associated weights as well. Finally, the uncoarsening phase, also known as the recovering and refining process, is completed by

returning to the graphs in the first phase to reconsider constituent nodes. During each recovery level, a local refinement optimization of the partition is applied to obtain more equal districts. Figure 5 visualizes the processes in the MLRB algorithm.

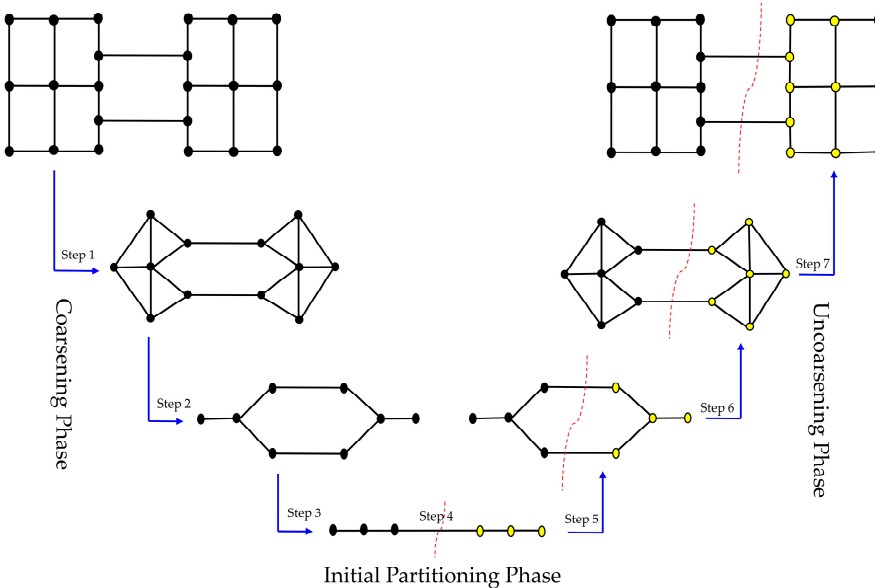

**Figure 5.** Different phases of a multi-level recursive bisection algorithm [72].

Alvisi [46] proposed a procedure for automated network sectorization using a combination of the MLRB graph partitioning algorithm and hydraulic simulation. However, unlike the traditional approach, it can simultaneously allocate the nodes to perform the best network partitioning into a given number of DMAs and identify the best locations for flow meters and valves in the network. Perelman et al. referred to a similar application [37] in which they used MLRB to compare performance in a WNP with other approaches, such as community structure and global clustering. This approach can assign weights to nodes and pipes that would otherwise not have been considered for the graph-clustering algorithm where the network clustering phase was considered a topology characteristic. Moreover, in terms of computational efficiency, MLRB showed advances in the uniform allocation of computation processes, considering more than one object simultaneously while minimizing the volume of information exchanged between them.

### 3.5. Spectral Graph Algorithms

Spectral graph theory is a mathematical approach to study the relationship of graph properties by associating both linear algebra and graph theory to determine the eigenvalue and eigenvector properties. Spectral clustering uses the spectrum in eigenvectors of the adjacent matrix to cluster groups of points into communities [49,54]. In spectral clustering, the row of eigenvectors of the Laplacian matrix for a pair of the nodes is similar if the nodes belong to the same cluster. Spectral-based graph clustering has been implemented in many fields over the last decade, especially in computer sciences, bioinformatics, and data analysis. Recently, in the field of WDN management, spectral graph theory has been used to define an optimal cluster configuration, a preliminary analysis of network vulnerability, and robustness through graph matrices eigenvalues [76,77] and a toolset for WDN management has been proposed [49]. Several types of research have applied spectral graph theory for WDN management, but in this subsection we focus on effective approaches that have been proposed for water network clustering. The core idea of spectral clustering is the Laplacian matrix as defined by

three equations. The first is the non-normalized Laplacian relationship, which solves a relaxed version of the Ratiocut problem proposed by Von Luxburg [78]:

$$L = D - A, \tag{4}$$

where **D** is a diagonal matrix of nodal degrees $k_i$, $\mathbf{D} = diag(\mathbf{d})$, in which $\mathbf{d} = [k_1, k_2, \ldots, k_n]^{\mathrm{T}}$. **A** is the adjacency matrix.

The other two matrices are normalized graph Laplacians, which are closely related and can be defined as

$$L_{sym} = D^{-1/2}LD^{-1/2} \text{ and} \tag{5}$$

$$L_{rw} = D^{-1}L, \tag{6}$$

where $L_{sym}$ is a symmetric matrix proposed to solve the NCut problem [78] and $L_{rw}$ is closely related to a random walk, which can be used to solve the same problem.

The Laplacian matrix of an undirected graph has the following properties [78,79]

- **L** is symmetric and positive-semidefinite with eigenvalues $\lambda_i \geq 0$ for all $i$.
- Every row sum and column sum of **L** equals zero.
- The smallest eigenvalue $\lambda_1$ of **L** equals zero.
- **L** has $n$ non-negative and the number of connected components in the graph equals the algebraic multiplicity of $\lambda_1 = 0 \leq \lambda_2 \leq \cdots \leq \lambda_n$.

Aim of spectral graph partitioning is to divide graph $G$ into $p \leq n$ subgraphs $G_1, G_2, \ldots, G_p$. Then,

$$V = V_1 \cup V_2 \cup \cdots \cup V_p \text{ where } V_i \cap V_j = \varnothing, \ i \neq j, \tag{7}$$

Let $G_k = (V_k, E_k)$ represents for subgraph $k$, in which $k = 1, \ldots, p$ and $V_k$ is the set of vertices of subgraph $G_k$. From Equation (7), an edge that has its endpoints in different vertex subsets is not contained in any of the formed subgraphs $G_k$ and is called an intercluster edge. Let a set of the intercluster edges with one endpoint in $V_k$ be denoted as Equation (8).

$$\partial(V_k) := \{ij : i \in V_k \text{ and } j \notin V_k\}, \tag{8}$$

Two different sets of edges can thus be distinguished as follows.

- intracluster edges: $E_1 \cup E_1 \cup \cdots \cup E_p$, and
- intercluster edges: $\partial(V_1) \cup \partial(V_2) \cup \cdots \cup (V_p)$.

From the optimal bipartitioning of a graph point of view, minimizing the *cut* values are objective functions. Von Luxburg [78] and Shi and Malik [80] proposed these functions to optimize *cut* value, called ratiocut method and normalized cut method, in Equations (9) and (10), respectively.

$$\min_{V_1, V_2, \ldots, V_p} \sum_{k=1}^{p} \frac{vol(\partial(V_k))}{|V_k|}, \tag{9}$$

$$\min_{V_1, V_2, \ldots, V_p} \sum_{k=1}^{p} \frac{vol(\partial(V_k))}{vol(V_k)}, \tag{10}$$

where $vol(\partial(V_k))$ is the sum of the weights on the all intercluster edges in $\partial(V_k)$; $|V_k|$ is the number of vertices in $V_k$; and $vol(V_k)$ is the sum of the weights on the vertices in $V_k$.

Equations (9) and (10) are NP-complete problems, however, they can be relaxed to find approximate solutions proved by Von Luxburg [78] and Shi and Malik [80] and reformed as Equations (11) and (12) following

$$\mathbf{LU} = \mathbf{U\Phi} \qquad \text{for ratio cut} \tag{11}$$

$$\mathbf{LU} = \mathbf{DU\Phi} \qquad \text{for normalized cut} \tag{12}$$

where $\mathbf{\Phi} := \mathrm{diag}\big(\lambda_1, \lambda_2, \ldots, \lambda_p\big) \in \mathbb{R}^{p \times p}$ and $\mathbf{U} := \big[u_1, u_2, \ldots, u_p\big] \in \mathbb{R}^{n \times p}$.

Equations (11) and (12) are eigenvalue problems for $p$ smallest eigenvalues $\lambda_1 = 0 \leq \lambda_2 \leq \cdots \leq \lambda_p$ of the Laplacian matrix $\mathbf{L}$ and their corresponding eigenvectors $u_1, u_2, \ldots, u_p$.

The spectral clustering algorithm for a non-normalized Laplacian matrix can be described as shown in Figure 6. For other normalized spectral clustering, refer to Reference [78].

Step 1: Construction of WDN as a graph $G = (V, E)$.
Step 2: Define the adjacency matrix $\mathbf{A}$ or weighted undirected adjacency matrix $\mathbf{W}$.
Step 3: Compute the Laplacian $L$ as Eq. (4).
Step 4: Compute the first $k$ eigenvectors $u_1, \ldots, u_k$ of $L$.
Step 5: Delineate matrix $\mathbf{U}$ containing clustering information and the first $k$ eigenvectors as columns.
Step 6: From the $\mathbf{U}$ matrix, the $k - mean$ algorithm is used to cluster nodes of the network into clusters $C_1, \ldots, C_k$ for each row.
Step 7: Continuity checking of the obtained clusters $C_k$. The links that have start nodes and end nodes belonging to different clusters are defined as the set of edge cuts, $N_{ec}$ (i.e., boundary pipes).

**Figure 6.** Flowchart of a non-normalized spectral clustering algorithm.

Using the spectral clustering methods mentioned above, many studies have adopted the spectral graph theory [47–49,81]. Di Nardo et al. [47] defined the optimal layout of DMAs in a real WDN. The authors took into account both geometric features (i.e., connectivity) and hydraulic pipe features (i.e., diameter, length, conductance, flow) through weight-adjacency matrices, which led to significantly different layouts of the DMAs. In particular, they compared different weighted spectral clustering (i.e., normalized versus non-normalized Laplacian) to determine the effectiveness of those approaches and the optimal choice of weights.

One of the most useful approaches for handling WDN complexity is a graph spectral technique (GST). Di Nardo et al. [49] pointed out that GST can analyze network topology by taking advantage of the properties of some graph matrices, providing a complete toolset to evaluate the performance and the evolution of networks. Based on two graph matrices (i.e., adjacency and Laplacian), the authors highlighted that GST metrics and the algorithms accomplish some crucial tasks of WDN management using topological and geometric information. In addition to the inherent ability to define the optimal clustering layout proposed in the literature, GST assisted in the calculation of a surrogate index for assessing topological WDN robustness using two indices, such as spectral gap and the algebraic connectivity. Additionally, the spectral technique also provides a framework that ranks important nodes in WDN to provide a useful approach to identify the location of valves or sensors or even determine the most influential nodes in a network [49].

Similarly, Liu and Han [48] also proposed a strategy for automatic DMA design based on spectral clustering and graph theory. The spectral algorithm was used to determine the best node clusters, which correspond to the DMAs' configuration based on steady-state simulations using the peak-hour demand. The study proposed a method for DMA design that combines spectral clustering, graph theory, and network centrality analysis. First, a combination of graph spectral theory and k-mean clustering was implemented to generate the initial DMAs. Then, to improve the cluster quality, a genetic algorithm (GA) was added to converge on a global optimum. To measure network centrality, the eigenvector centrality [82] was used to identify the critical nodes, and edges betweenness centrality [83] were adopted to measure the important pipes, creating a high-quality cluster. Most recently, Zevnik et al. [81,84] proposed spectral graph partitioning based on a generalized and normalized cut method and compared it with two known spectral methods (ratio cut and normalized cut).

In the field of machine learning, graph Laplacians are used not only for clustering, but also for many other tasks, such as semi-supervised learning. Herrera et al. [52,85] demonstrated that graph-based semi-supervised learning methods [86] can take into account various criteria for segmentation of WDNs into DMAs. In this method, the kernel matrix [87] was first defined and then the adjacency matrix was enriched by adding hydraulic data such as weight factors to transform the results into a kernel matrix. The spectral clustering algorithm was adapted to this new matrix. Finally, graph-based semi-supervised learning methods were conducted. A similar method was found in a study by Giudicianni et al. [36,88] in which semi-supervised multiscale clustering was used to create dynamic DMAs. Compared with methods that use only topological connectivity or vector information, semi-supervised clustering showed improvement by integrating both forms, leading to the efficient development of robust DMAs.

Spectral clustering can take into account topological, geometrical, or hydraulic aspects as weight factors, which allows for a careful consideration of alternative factors that can affect the goals of the DMA design and provide a multidimensional view to help managers make better decisions. However, for large-scale networks that have thousands of nodes and links, spectral clustering has limited applicability and becomes infeasible due to the computational complexity of $Q(n^3)$ [52], where $n$ is the number of nodes.

### 3.6. Multi-Agent Approach

A multi-agent system (MAS) [89] is a loosely coupled network of autonomous problem solvers composed of multiple interacting intelligent agents. Each agent works independently but can also interact with others to solve potential conflicts through negotiation. The properties of MAS can be described as follows.

1.  Each agent has an imperfect standard or may lack the capacity for problem solving, and therefore has a somewhat limited and unbalanced perspective;
2.  There is no global information;
3.  Data is decentralized; and
4.  Computation is asynchronous.

MAS networks are suitable for handling multiple-problem approaches or multiple-agent solving entities. Known as a complex system due to the joining of many physical devices, a WDN comprises multiple parties with different goals, actions, and information and is a dynamic system. A small change in behavior of the parties may result in unpredictable patterns in the entire system. WDNs and multi-agents exhibit a strong similarity, and MASs can therefore provide solutions to distributed applications, such as the problem of network partitioning, which is known to be complex and has multiple constraints. MASs have been successfully applied to heterogeneity problems in the water field. They have proven to be highly efficient at optimizing water networks, control systems for municipal water, water pollution diagnosis, water quality enhancement, and water demand management [90].

In terms of WDN clustering, many elements must be simultaneously considered. A network can be divided into elements, which are considered as agents that communicate with each other. Izquierdo et al. [90] were the first to develop a suitable software environment to formulate DMA segmentation in a WDN using a multi-agent approach. They proposed a likelihood method by running a simulation as verification to divide networks into subsectors based on sources, nodes, and pipe properties, which consider nodes and pipes as agents of a separate breed. This can be seen as a premise to improve as well as implement the multi-agent methods in different studies. Herrera et al. [52] assumed an a priori set of DMAs based on the homogeneity of the districts, which was related to the source tanks in the network, where each reservoir was seeded for the corresponding DMAs. These agents adopted a method of clustering by elicitation, linking their adjacent nodes to the source points, and scanning the likelihood of each being assimilated into the corresponding DMA.

On the other side, Hajebi et al. [51] combined a *k*-means clustering method and multi-agent approach to WNP. In particular, *k*-means graph clustering was used to divide the network topology into

a predetermined number of clusters and then a MAS was implemented to negotiate the configuration of the network by adjusting nodes on the boundary pipes of the corresponding clusters while considering the hydraulic constraints. Compared with previous studies by Herrera et al. [50], differences in the approaches are evident. In the former study, DMA layouts were determined based on the source points of the network and expanded by negotiation, while the latter started from the geographical clustering of the network and boundary pipes were modified to obtain the best hydraulic performance.

## 4. Sectorization to Locate Flow Meters and Valves

Immediately after forming DMAs from a clustering phase, it is important to optimize the state of the boundary pipes, namely the position and number of gate valves and flow meters required to achieve reliable DMA operation. This is also known as a decision support tool to help utilities solve optimization problems while investigating the best trade-offs between the sectorization cost of investment versus indicators of the benefit of DMA installation. The position of these devices is important because closing a valve impedes hydraulic behavior and reduces network reliability. After segmentation of a network into districts, the standard requirement of a network is still to ensure adequate quantity and quality as well as appropriate pressure. Many algorithms and heuristic procedures have been proposed to find the optimal solution for this phase, which is concerned primarily with optimizing hydraulic performance and leak-reduction efficiency.

Many heuristic procedures are available to maximize the benefits of physical demarcation of a water network into DMAs, but this work has been implemented largely with evolutionary algorithms, which include single or multi-objective functions and are constrained by hydraulic or economic conditions. In this section, we focused on the optimization approaches to discuss the sectorization phase in WNP.

### 4.1. Single-Objective Optimization Approach

After a set of boundary pipes $N_{bp}$ are defined in the clustering phase, the first objective is to determine how many flow meters $N_{fm}$ and gate valves $N_{gv}$ to insert along the boundary pipes. Most researchers agree that fewer flow meters will reduce reconstruction and operating costs, as well as the initial cost of installing the flow meters, which are often more expensive than gate valves [6,17,47,49].

In addition, the positions of the gate valves and flow meters have a significant effect on network properties such as hydraulic performance, resilience index, leakage rate, and water quality. WDN sectorization should therefore be considered as a multi-objective optimization problem to maximize the benefits of implementing DMAs. However, to simplify computational demands, some hypotheses or heuristic processes have been proposed to convert a multi-objective problem into a single objective and apply evolutionary algorithms to achieve feasible or optimal solutions.

Because the number of feasible solutions is large, various heuristic optimization techniques have been studied [6,91–94]. Although the objective functions and constraints are different among the various approaches, they all aimed to achieve as high as possible a network performance after sectorization. The total power of a WDN is classified into the dissipated power at pipes (i.e., internal power loss) and the supplied power at node (i.e., external power supplied). Di Nardo et al. [6,17,18,49,92,95,96] suggested the objective function to maintain the hydraulic performance of the network at the lowest dissipated power that consequently maximizes the nodal supplied power by maintaining the nodal head as high as possible after sectorization. The objective was defined in the following equation.

$$max\left(\gamma \sum_{i}^{n}(z_i + h_i)Q_i\right),\tag{13}$$

where $\gamma$ is the specific weight of water and $z_i$, $h_i$, and $Q_i$ are the elevation, pressure, and water demand at node $i$, respectively. For a large and complex network, it is not easy to decide how many flow meters should be positioned among boundary pipes due to the trade-offs between hydraulic

performance and investment cost. To deal with this problem, Shao et al. [97] proposed a function that converted a dual-objective problem (i.e., hydraulic performance and cost) to a single-objective problem by considering the master-subordinate relationship of the two objective functions, which improved the computational efficiency.

In addition, a changing flow due to pipe failure can cause changes in velocity, energy losses in pipes, and pressure at nodes, especially in a looped network. This will cause changes in the pathway of water particles to the nodes. Moreover, if a node is being supplied at the minimum required pressure, it will not be able to provide the necessary flow and pressure. In the worst-case scenario, the network must ensure a capacity to provide a surplus power to overcome system failures. This is an approach proposed by Todini [98] to measure system resilience when redesigning a system or when system malfunction occurs. Based on that criterion, when reconstructing the system by creating isolated DMAs, several studies [34,36,46,60] have used a resilience index (Equation (14)) as an objective function for sectorization optimization. The objective function can be maximized to indicate that a greater surplus of available power leads to a higher network resilience such that:

$$I_r = \frac{\sum_{i=1}^{n_n} Q_i (h_i - h_{min})}{\sum_{r=1}^{n_r} Q_r H_r - \sum_{i=1}^{n_n} Q_i h_{min}},$$  (14)

where $n_n$ and $n_r$ are the numbers of demand nodes and reservoirs, respectively; $Q_i$ and $h_i$ are water demand and pressure at node $i$; $Q_r$ and $H_r$ are the water discharge and total head of the source or tank r; and $h_{min}$ is the minimum required pressure for adequate service.

For cost analysis, Gomes et al. [99] proposed an optimization model to design DMAs based on different decision-makers' options to reduce the total cost. Referring to different future scenarios for water demand (that will increase) and the infrastructure degradation forecasts, the cost of WNP was assessed. An objective function aims to minimize the cost of DMA redesign and first considers the cost of pipe reinforcement or replacement with flow meters and gate valves. Second, to ensure that the model approximates reality, the cost function is multiplied by the weight or probability of occurrence for each of the scenarios. A similar study that considers economic and energy criteria for DMA design can be found in Di Nardo et al. [75].

For reducing the leakage in WDNs, Creaco and Haidar [100] proposed a linear programming framework to optimize control valve settings. Accordingly, isolation valve closures, control valve installations, and DMAs creation are simultaneously optimized to search optimal solutions in the trade-off between installation costs, leakage, and demand uniformity across DMAs.

To solve the optimization problems mentioned above, evolutionary search algorithms have been applied. The GA [101] has been widely implemented by Di Nardo et al. [6,47,75,91,95]. Meanwhile, Shao et al. [97] improved the GA for faster and superior layout of flow meters and valves by modifying crossover and mutation mechanisms. In addition, the simulated annealing algorithm was presented by Gomes et al. [61,99,102].

*4.2. Multiple-Objective Optimization Approach*

WNP is a complicated task and must achieve many goals. Zhang et al. [42] proposed a multi-objective optimizing approach for sectorization, in which three objective functions were used: the number of boundary pipes, network pressure uniformity, and water age uniformity. Zhang et al. [103] proposed a multi-objective optimization to obtain more reasonable schemes for sectorization of a WDN by simultaneously considering pressure stability, water quality safety, and system reconstruction costs. For pressure stability, the average pressure was minimized, but was still above the minimum pressure. Water age is the time spent by a water parcel as it travels from a source to nodes in the network, which represents the water quality in a WDN. Additionally, the costs of installing flow meters and valves should be minimized depending on the size and number of DMAs. However, if only the initial investment cost of these devices is considered, it is impossible to comprehensively evaluate the

expense of the WNP. De Paola et al. [104] presented an objective function to deal with the total cost of sectorization, which also involves water leakage costs and energy consumption by pump operations.

Even if we consider all the criteria in the process of network sectorization, it is impossible to provide an optimal result due to trade-offs. The current partitioning techniques prioritize only a few representative sets of criteria, and do not fully address the best practical problems of DMA design. In an attempt to provide a comprehensive review of the criteria when dividing the network as close as possible to reality, Hajebi et al. [105] considered two sets of objectives in the sectorization task, the structural objective and hydraulic objective. For the structural objective, they considered the minimum cut size and minimum boundary pipe diameter. For the hydraulic objective of the network after segmentation, they considered four objectives, including minimization of the average excessive pressure at nodes, minimization of dissipated power, minimization of elevation differences in each DMA, and maximization of network resilience.

In another combination, a series of energy, operative, and economic criteria were optimized in the sectorization process [58,59]. Five objectives were addressed concerning the minimum deviation of the resilience index [95], which measures the capacity of the network to conquer system failures, the ability of the system to ensure an appropriate service pressure in the whole network, and minimization of the variation of the operational power, which assessed the reliability of a sectorization layout based on a pressure target. Operative criteria were also formulated as objective functions of pressure at nodes. When pressure dropped, a reduction in the leakage was expected. Variation of nodal pressure should therefore be minimized. However, pressure at nodes after sectorization needs to be higher than the minimum threshold required for service. To accommodate this constraint, a penalty cost for a nodal pressure deficit was added. Finally, the cost criteria when performing sectorization are also important. The cost of positioning and operation of flow meters are expected to be higher than that for boundary valves. Therefore, an objective function to minimize the cost for installing and operating the devices needs to be considered. Similarly, Brentan et al. [43] adopted a multilevel optimization concept to reduce the complexity of sectorization. In their approach, two groups of the objectives were minimized. The first one corresponded to structural costs, which were related to valve and flow meter installation, while the second group reflected hydraulic performance, such as minimum pressure and maximum resilience index.

Giudicianni et al. [88] recently developed a heuristic framework for dynamic partitioning of WDNs using multi-objective functions to address different goals for saving energy, water, and costs. Specifically, they proposed a method for zero-net energy management of a WDN using microhydropower stations [106] along the boundary pipes during the day and a reduction of water leakage at night.

To provide a comprehensive method for optimal DMA design, Galdiero et al. [107] proposed a decision-support tool that focused on water network segmentation by considering two objective functions. A total cost function including the initial cost for device investment and a daily cost due to water leakages were considered to minimize and compared as trade-offs with changes in hydraulic performance in terms of the maximum resilience index. To integrate different algorithms and multi-objective functions to the development of a decision support tool, Di Nardo et al. [18] developed advanced software called SWANP. A clustering model was implemented based on MLRBs, which are multiagent approaches to water network clustering. In the sectorization phase, an optimization algorithm was proposed using multi-objective functions to find optimal DMA configurations that complied with the level of customer service and considered the minimum pressure and maximum resilience index, and balanced the cost of investment and operation by minimum devices inserted to achieve isolated DMAs. SWANP was written on a Python environment with a user interface and was evaluated as an effective decision support system providing the manager with different optimal layout solutions.

Many optimization algorithms have been applied to deal with discrete nonlinear combinations and solve the multi-objective optimization of water network sectorization. NSGA-II [74] has been

widely applied to multi-objective optimization problems. In terms of water network sectorization, NSGA-II has been used in many studies [103–105,107] to obtain the Pareto front, which contains a set of Pareto optimal solutions, thus providing support for managers charged with making more accurate and reasonable decisions based on their priorities and objectives. Zhang et al. [42] implemented an auto-adaptive many-objective algorithm [108] to solve the sectorization problem that shows some new features compared with NSGA-II. Giustolisi and Ridolfi [41] used a multi-objective GA to support network segmentation. Campbell et al. [58], and Gilbert et al. [94] applied an agent-swarm optimization algorithm [109]. In addition, the combination of three optimization algorithms of GA, particle-swarm optimization [110], and soccer-league competition [111] was suggested in Brentan et al. [44].

### 4.3. Iterative Approach

In addition to the described optimization methods, iterative methods were applied to the placement of flow meters and valves [19,41,48,96]. An iterative method is a mathematical procedure that can generate a feasible solution using an initial guess to generate a sequence of solutions. The result is considered convergent when the initial set of criteria is met. Diao et al. [19] considered DMA size and minimum pressure as criteria, and used them as constraints in the heuristic-based iterative method to define the feedlines for each DMA. The approach determines the location of flow meters among boundary pipes between DMAs. In addition, Liu and Han [48] proposed an iterative method based on a heuristic procedure to determine the best location of flow meters subject to constrain head pressure at nodes. The iterative method permits the selection of one flow meter based on the shortest path from the source that can improve the pressure in each iteration. Di Nardo and Di Natale [96] inserted a certain number of flow meters on boundary pipes and then designed a procedure to alternately change the quantity and position of flow meters to achieve an optimal solution based on hydraulic performance constraints testing.

### 4.4. Adaptive Sectorization for Dynamic DMAs

In normal working conditions, a DMA layout is permanent and optimized by WNP processes to satisfy the hydraulic constraints and network performance indices. In abnormal cases, such as pipe breaks, fire-fighting, and unexpected increases in water demand, permanent DMAs may produce failures in preserving or maintaining regular water supplies. To adapt to such conditions and overcome the drawbacks that a permanent DMA can cause, Giudicianni et al. [36,88] proposed creating dynamic DMAs that allow for expansion of existing DMAs. That is, the small DMAs are dynamically aggregated into larger ones using a semi-supervised clustering algorithm. This approach allows for a new configuration that always includes former DMAs and maintains the set of boundary pipes at each subzone. In some cases, by controlling the dynamic gate valves, the operator restores connectivity to its original configuration and consequently helps the utility periodically desegregate.

In addition, Wright et al. [112] proposed a method of integrating the advantages of DMAs in reducing leakage while improving network resilience and water quality by dynamically reconfiguring network topology and pressure control through optimizing valve settings and boundary pipes status using a sequential convex programming approach. The proposed approach leaned on the self-powered multifunction network controllers that allowed adjustments of the network topology and continuously monitored the dynamic hydraulics based on consumers' actions (i.e., the varieties of the system's water demands). In low demand periods, original DMAs were preserved to capture the minimum night flow within small isolated areas and maximize the ability to detect leaks. In peak demand periods, DMAs were then aggregated into larger pressure-controlled zones to maximize the resilience index and improve energy efficiency due to reduced internal losses that come with using larger DMAs. The core idea here was inserting the network controller associated with dynamically reconfigurable DMAs that allows a utility to monitor high-resolution, time-synchronized, dynamic pressure conditions of the network. Similarly, Perelman et al. [113] used a linear programming approach to automate reconfiguration of an existing WDN into DMAs. The network was reorganized into a star-like topology

by identifying and decomposing the existing network into a main network and subnetworks based on graph theory. Center nodes were located in main pipes and played an important role as key connections between transmission main pipes with water sources and other nodes in the rest of the network. The proposed method provided a flexible tool for water utilities by allowing only existing valves to be closed, saving investment and operation costs for additional valve installations.

Ideas derived from DMAs' limitations in emergencies, especially in the case of fire-fighting, overcome this drawback. Di Nardo et al. [18,114] recently proposed a method that allows for the redesigning of static DMAs to dynamic layouts. A heuristic procedure based on a GA was developed to determine the number of gate valves that have to be motorized and remotely controlled to satisfy hydraulic performance in a fire-fighting event. This practical technique provided system operators with a quick decision-making tool to respond to unexpected incidents in the network and eventually leads to a smart water management paradigm. Unlike the approaches proposed above, Santonastaso et al. [115] developed a dynamic scheme for adjusting a WNP by accounting for the real positions of isolation valves present in the WDN. To do this, the adjacency matrix of the WDN was changed and replaced with a dual topology based on WDN sectorization and isolation valves. DMAs obtained in this approach allowed topology matrix segments to merge while inter-DMA boundary pipes were forced to be selected among the valve-fitted pipes that separated segments. Feasible DMAs were generated that did not require additional isolation valves.

To visualize the procedure of WNP technique in the abovementioned different phases, Figure 7 illustrates the procedure of WNP for a real-life WDN in Parete town, South Italy [47,49]. In this case, 4 DMAs were generated in the clustering phase based on the normalized spectral algorithm, and a heuristic procedure based on GA is applied in sectorization phase to locate the control devices while maximizing the total nodal power of the network.

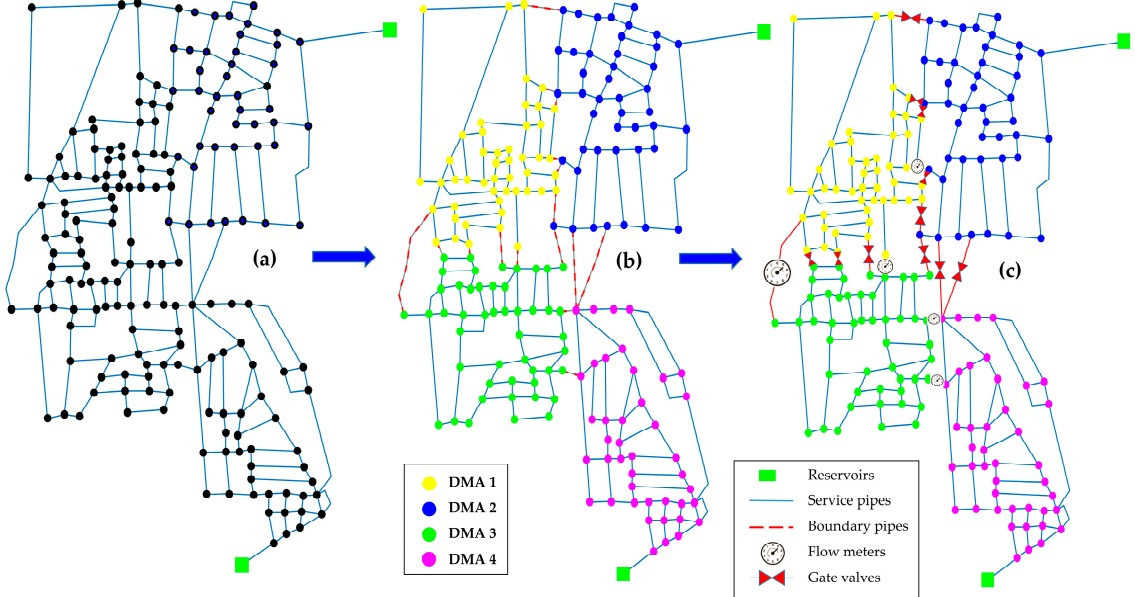

**Figure 7.** Illustration of the water network partitioning (WNP) procedure implemented to a water distribution network (WDN) in Parete, South Italy (adapted from Di Nardo et al. [47,49]); (**a**) original water network, (**b**) clustering phase, and (**c**) sectorization phase.

## 5. Performance Assessment of Water Network Partitioning

As mentioned in Section 2, the water network segmentation to DMAs is expected to bring many benefits along with effective reduction measures for invisible water losses, manage pressure uniformity, and prevent network contamination. However, in some cases, it may also decrease the hydraulic performance and reliability of the network. To measure how this change affects network

hydraulic behavior, performance indices (PIs) can quantify the benefits and drawbacks that DMAs bring. A PI test allows for the evaluations of the performance of the original networks compared with those of the divided networks. Most were estimated using a hydraulic simulation solver based on the demand-driven analysis. Most studies applied multiple PIs to evaluate the effectiveness of DMAs, such as resilience indices, pressure indices, uniformity indices, water quality indices, and fire protection indices.

The first to be mentioned is the resilience metric, which monitors the power balance of a water network, as proposed by Todini [98] in the form of Equation (14). According to this metric, WDN resilience is defined as the capacity to overcome sudden system (hydraulic or mechanical) failure. The resilience index is often used to evaluate the performance of a WNP as a comparison of network power before and after the sectorization. Most studies have stated that the resilience index is not significantly affected by network sectorization compared to its benefits [3]. Herrera et al. [116] proposed a graph-theoretic approach by adopting the K-shortest paths algorithm [117] to assess the resilience of larger-scale partitioned WDNs. To do this, all nodes in every DMA are aggregated into a sector-node, where a new DMA-graph is represented by sector-nodes and edges that are abstracted by sector-to-sector connectivity. A mapping function was used to transform the resilience of nodes to a sector-scale resilience. They showed that the resilience of individual nodes in the DMAs closely links to the corresponding sector-nodes resilience. This establishes a different way to identify DMA configurations that have a major impact on the resilience index.

If a resilience index evaluates the overall performance of the network, hydraulic statistical indices allow for the evaluation of the level of service that a water system supplies to its customers, providing managers information on pressure change in terms of mean, minimum, maximum, and spatio-temporal deviations. More specifically, Di Nardo et al. [118] proposed several indices, such as a mean pressure surplus and mean pressure deficit compared with the design pressure.

On the other hand, water quality is measured by water age in a network and is influenced by network topology, flow velocities, and pipe lengths. The age of the water affects residual chlorine levels. Lower chlorine induces bacterial growth, and higher values indicate worse performance. Many studies used the water age index as an indicator to assess the impact of DMA design on water quality. Grayman et al. [10] and a series of studies by Di Nardo et al. [118] illustrate that after incorporating DMAs into a WDN, there was no systematic difference in the computed average water age between alternative scenarios. Although there can be significant variations in water age by node due to valve closures, when considered as a whole, no homogeneous difference was found.

Meanwhile, partitioning a WDN into subnetworks with gate valves can prevent the spread of contamination in the case of malicious attacks. Di Nardo et al. [28,119] proposed a method that uses a simple backflow attack with cyanide to investigate the effects of network partitioning. Grayman et al. [10] proposed an index to quantify the potential health impacts from contamination incidents in the WDN.

Several hydraulic uniformity indices [44,48,120,121] have been developed to evaluate the performance of DMAs. The size uniformity index reflects the cumulative demand deviation of all DMAs compared with a hypothetical DMA with average demand, for which a smaller value indicates a better performance. Pressure uniformity was suggested to guarantee that all nodes belonging to a certain district would have similar pressure patterns. A lower index value indicates better performance. Total head uniformity is also used to measure the variance of total heads along the nodes, which has the same meaning as pressure uniformity. Liu and Han [48] proposed a decision-making framework to determine the optimal DMA design by quantifying various indices, such as DMA uniformity, modularity index, and resilience index. Similarly, evaluating the benefits brought by DMAs in terms of cost-benefit analysis allows managers to make sensible decisions and create functional and efficient DMAs. Ferrari and Savic [25] proposed a comprehensive method that considers alternative DMA configurations to show the savings that utilities can obtain by considering three indices related leakage reduction, burst-frequency reduction, and pressure-sensitive demand

reduction compared with the original network. Pressure reduction across the network was the main factor leading to reducing leakage and burst frequency. The study provided a decision-support tool for economic performance analysis of various DMA layouts.

In the case of a fire, while water demand is high for fire-fighting at a few nodes, the network must still have the capacity to supply enough water to users, especially during peak demand hours. This superposition of demand creates energy-loss leaps in pipes, leading to lower pressures at that time. Moreover, when creating isolated DMAs, some pipes feeding a district are closed and this could have negative impacts on the amount of flow entering a DMA. To test this situation, Grayman et al. [10] and Di Nardo et al. [118] developed a fire protection index based on the number of nodes with a pressure lower than the required pressure designed for the fire-fighting event. Those results indicated that some negative pressure values were occurring while most of the nodes had acceptable pressure. However, a significant difference was found between looped and branched networks.

WDN is a dynamic system in which pressure can vary significantly due to variations in water demand at nodes. Addressing spatial-temporal variability of water demand in the network, Di Nardo et al. [122] proposed a procedure for WNP under stochastic water demand and quantified its effects on hydraulic performance. The study revealed that by applying random variability of water demand, the magnitude of pressure distribution within the network was affected significantly. This led to a decrement of surplus pressure and network resilience compared with the constant-demand condition. To create feasible DMAs, especially for a WDN characterized by a small deviation between the surplus pressure and required standard pressure, spatial-temporal variability of water demand should be considered in WNP.

## 6. Discussion and Future Work

This paper provided a comprehensive review of the relevant studies on WNP over the last decade. The WNP procedure consists of two basic phases. First, the clustering phase involves the formation of the sizes and dimensions of DMAs as well as the definition of the boundary pipes that feed or interconnect DMAs. This phase is commonly associated with use of a clustering algorithm. In this study, six commonly applied algorithms such as (i) Graph theory, (ii) community structure algorithm, (iii) modularity-based algorithm, (iv) multilevel graph partitioning, (v) spectral graph algorithm, and (vi) multi-agent approach, are presented and discussed in-depth to understand how they work and handle in formation of the feasible DMA configurations. These algorithms are commonly based on the graph theorem that relies primarily on the network's topology. Since WDN is an infrastructure system with particular properties, the water network clustering algorithm allows for tailoring by appending weights to pipes or/and nodes to mimic distributing loads across the WDN. Many criteria for DMA design, such as hydraulic performance, network topology, system reliability, water quality, and cost-benefit ratio, are considered in this phase to minimize the number of boundary pipes and its goal is to define the reasonable size and configuration of DMAs.

Second, the sectorization phase is a physical segmentation process that identifies the position of gate valves and flow meters among the set of boundary pipes to satisfy operational constraints. This task requires the designer to apply an optimization algorithm or heuristic procedures to ensure that the locations of devices will have the least negative impact on the hydraulic performance of the network, minimize the energy use and leakage and be cost-effective.

The improvements and innovations in WNP developed to date often come from innovative approaches, either with the clustering algorithm or sectorization optimization. Those innovative features have emerged from combinations of clustering algorithms and alternating with flexibility or/and developing various objective functions based on the different criteria designs to propose a heuristic procedure for creating the most reliable DMAs. Many different approaches have been proposed for the automated creation of DMAs. However, several shortcomings remain to pursue in the future.

- Clustering is the crucial phase for WNP. Several algorithms and software tools were developed to deal with the large-scale networks that are burdensome to tackle manually. Various engineering aspects were embedded as weights to modulate WDN characteristics. More extensions of the existing graph clustering algorithms to weighted networks would be of great interest, as well as novel methods for clustering directed graphs.

- While there were many different approaches for the identification of DMAs in water networks, few studies tackled to determine the optimal number of DMAs for a given network. It is an open question and requires a decision-making procedure utilizing various network performance quantification metrics.

- In the sectorization phase, it still lacks how to assess the pump and tank operations in the partitioned network. Moreover, an approach to consider the consequences of device placements to the leakage, energy use, and post-damage restoration should be studied quantitatively in this phase.

- A demand-driven analysis (DDA) is generally used for WNP under the normal working condition at peak hour demand. In DDA, the supplied demand is assumed to be independent of pressure and this approach is valid when the pressure is above the minimum pressure requirement. In reality, a WDN works more likely as a pressure-driven analysis (PDA), in which the nodal consumption depends on the nodal pressure. Therefore, in pressure-deficient conditions (e.g., pipe failures, fire-fighting, unexpected water demand increase), a PDA should be applied for the novel dynamic WNP that adapts flexibly under the abnormal operating conditions.

- Last but not least, a WDN is supplied by single or multiple sources, with different elevations, divergent intended pressure in each zone. It also can be expanded or replaced according to urban planning needs. Further research should address the change of network's topology, controlling hydraulic uniformity in each zone as well as improving system resilience. Future research needs to be conducted to improve the abovementioned limits and eventually to provide optimal DMA layouts for efficient network operation and management.

**Author Contributions:** Conceptualization, X.K.B.; writing—original draft preparation, X.K.B.; writing—review and editing, D.K. and M.S.M.; supervision, D.K. All authors have read and agreed to the published version of the manuscript.

**Funding:** This research was supported by the EDISON* Program through the National Research Foundation of Korea (NRF) funded by the Ministry of Science, ICT & Future Planning (Grant No. 2017M3C 1A6075016). * Education-research Integration through Simulation On the Net.

**Conflicts of Interest:** The authors declare no conflict of interest.

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
