# Peer review of "Water Network Partitioning into District Metered Areas: A State-Of-The-Art Review"

_water, doi:10.3390/w12041002_

Round 1

Reviewer 1 Report

Dear Editor

The paper is an interesting review about the design, the advantages and the drawbacks of water network partitioning, that represents one of the most useful strategies for the management of water distribution systems. The scientific community lacked such a work that, as a consequence, represents a crucial step in the water engineering research field. The paper is well-written (only some parts are a bit confused) and it lists the most important works done in such a field, also including very recent papers. However, some major revisions are required that I would like to see addressed before the acceptance of the paper. Listed are my revisions that I recommend to consider; often, I also suggested to add and comment some papers that can clarify some of the aspects that the Authors dealt with:

  • In the Introduction, I suggest to enlarge the set of the benefits of the water network partitioning and refer to these papers:
  • Ilaya-Ayza, A. E., Martins, C., Campbell, E., & Izquierdo, J. (2017). Implementation of DMAs in intermittent water supply networks based on equity criteria. Water, 9(11), 851.
  • Ciaponi, C., Creaco, E., Di Nardo, A., Di Natale, M., Giudicianni, C., Musmarra, D., & Santonastaso, G. F. (2019). Reducing impacts of contamination in water distribution networks: a combined strategy based on network partitioning and installation of water quality sensors. Water, 11(6), 1315.
  • Line 94: “It is reported to be as low as 3–7% in a well-maintained system in the Netherlands and as high as 50% in some undeveloped countries”, I suggest to refer to the following papers:
  • Beuken, R. H. S., Lavooij, C. S. W., Bosch, A., & Schaap, P. G. (2008). Low leakage in the Netherlands confirmed. In Water Distribution Systems Analysis Symposium 2006 (pp. 1-8).
  • Lambert, A. O. (2002). Water Losses Management-International Report: Water losses management and techniques. Water Science and Technology-Water Supply, 2(4), 1-20.
  • The Authors can refer to the following paper, when they deal with the proper number of DMAs (page 5) and/or the possibility to model WDNs as a graph (page 6):
  • Giudicianni, C., Di Nardo, A., Di Natale, M., Greco, R., Santonastaso, G. F., & Scala, A. (2018). Topological taxonomy of water distribution networks. Water, 10(4), 444.
  • In the bullet point at page 5, I suggest to mention also the importance to take into account the topology of the network which affects the clustering phase and the application of the clustering algorithms;
  • Check typos in the paper, such as line 544 (an MAS), line 614 (in by Di Nardo);
  • Line 316: for the sake of consistency, I suggest to change the symbol L with m, since previously m has been used to indicate the total number of edges in the network (line 224);
  • Line 456: I suggest to better define the D matrix, reporting its mathematical definition, as well as, referring to the fact that it is a diagonal matrix;
  • Equation 7: could you specify what does it represent in hydraulics?
  • Paragraph 3: I suggest to divide this part in six subsections, like the number of algorithms that you described;
  • Subsection 3.1: since in the paper you often write about weighted graphs, I suggest to extend the description also from a mathematical point of view, like you did for unweighted graphs;
  • Figure 2: I suggest to add the steps for the choice of the weights (according to the specific goal) and the number of DMAs (when possible, according to the type of clustering algorithm and in accordance with guidelines);
  • Line 363: could you better explain the difference between “Global clustering, community structure, and graph partition”? According to that, I suggest to refer:
  • Fortunato, S. (2010). Community detection in graphs. Physics reports, 486(3-5), 75-174.
  • Page 10: the Authors deal with the comparison of different algorithms for defining the shape and size of DMAs; I suggest to also refer to the following paper that compares the performance of the graph partitioning and the spectral clustering for WNP, since both of them are designated in the paper as among the major algorithms for the clustering of WDNs:
  • Di Nardo, A., Di Natale, M., Giudicianni, C., Greco, R., & Santonastaso, G. F. (2017, November). Water distribution network clustering: Graph partitioning or spectral algorithms?. In International Conference on Complex Networks and their Applications (pp. 1197-1209). Springer, Cham.
  • Subsection 3.4: I suggest to better clarify the part regarding the Laplacian matrix and the possibility to relax the cut-problem and according to that, I suggest to refer:
  • Shi, J., Malik, J.: Normalized cuts and image segmentation. IEEE Trans. Pattern Anal. Mach. Intell. 22, 888–905 (2000).
  • Paragraph 5: I suggest to also refer to the following paper that deals with the framework for assessing the performance of water distribution networks after partitioning:
  • Herrera, M., Abraham, E., & Stoianov, I. (2016). A graph-theoretic framework for assessing the resilience of sectorised water distribution networks. Water Resources Management, 30(5), 1685-1699.
  • Line 826: “However, for an infrastructure system with particular properties, the water network clustering algorithm allows for tailoring by appending weights to pipes or/and nodes to mimic an asymptotic area (the WDN).”. could you better clarify this sentence? What does “asymptotic area” mean?
  • In the conclusion, the paper missed proper future directions; I suggest to better highlight this part, since it represents a crucial point for a review. In this regard, it would be better to provide bullet points regarding the still open tasks to address in the field of water network partitioning. In the following some advices to make this section clearer. In line 842 the Authors stated: “In the clustering phase, most approaches use graph theory, which only analyzes a network based on connectivity without considering the hydraulic characteristics of WDNs.” that is in contrast with the sentence in line 827“the water network clustering algorithm allows for tailoring by appending weights to pipes or/and nodes” and also with the fact that it is possible to adopt the weighted graphs to take into account the hydraulic/geometric characteristics. Refer also to:
  • Di Nardo, M. Di Natale, C. Giudicianni, G.F. Santonastaso, V. Tzatchkov, J.M.R. Varela, V.H.A. Yamanaka, (2016). Water supply network partitioning based on simultaneous cost and energy optimization. Procedia Engineering 162 ( 2016 ) 238 – 245.

I suggest to better clarify these aspects. Still, “Moreover, the algorithms have been validated on small networks only.”, I suggest to rewrite this part and also refer to:

  • Gilbert, E. Abraham, I. Montalvo, O. Piller, (2017). Iterative Multistage Method for a Large Water Network Sectorization into DMAs under Multiple Design Objectives. Journal of Water Resources Planning and Management, 143(11), 04017067.

Moreover, could you better clarify this sentence in line 845: “but no objective function related to the intrinsic hydraulics of the network has been identified”. Also this part is confused and it is not clear what the Authors want to point out. In line 848: “A WDN can be mapped onto a graph, but its characteristics differ with those of other networks. Because it was designed to transport and distribute water to users, a WDN has hydraulic properties represented by water demand, elevation and head pressure at nodes and the length, diameter, velocity, and flow at pipes. Some of these properties can change in time and space.”. If you want to stress the importance of considering the spatio-temporal variability of the characteristics of the WDNs, you could refer to the novel dynamic approaches that you listed in subsection 4.4 and eventually highlight new needs. 

  • Finally, for a review paper, I expected more Figures regarding examples of applications to real cases of the different procedures adopted in literature and listed in the paper. I really suggest to add pictures also of the devices generally used for the water network partitioning, in order to make the work more appealing for researchers and practitioners.

Author Response

Dear reviewer,

We appreciate your thoughtful comments and suggestions that helped us to improve the quality of our manuscript. A detailed description on how to address these comments is provided in the attached file.

Best regards,

Authors

Reviewer 2 Report

While I liked the paper overall, I have some comments that the Authors can benefit from. Hereinafter follow my major and minor comments:

Major comments

a - The paper is complete but two aspects could be inserted:

1 - boundary pipes are suitable sites for installing water quality sensors to detect contaminations, offering benefits in terms of cost and management (see paper entitled "Reducing impacts of contamination in water distribution networks: A combined strategy based on network partitioning and installation ofwater quality sensors"

2 - in some cases, the closure of isolation valves, the installation of control valves and the creation of DMAs are combined in a single optimization problem (see the paper entitled "Multiobjective Optimization of Control Valve Installation and DMA Creation for Reducing Leakage in Water Distribution Networks")

b - I think that merits and drawbacks of the works described in the paper should be commented on more. At the moment, only a few lines at the end of the paper exist.

Minor comments

a - line 96. I would search for another term instead of "commercial leaks". Please, refer to the terms adopted by the IWA task force on leakage in WDNs.

b - line 103. It's false that leakage is proportional to pressure. This is clear in the typical emitter equation, in which an exponent appears in pressure head, and also in the FAVAD equation.

c - line 158. I would speak about system's reliability (or redundancy) instead of resilience, in this case.

d - line 243. Do we need to partition WDNs to understand which nodes are disconnected from the sources. Please, explain better.

Author Response

(The authors gave the same response as above.)

Round 2

Reviewer 1 Report

The author followed all review suggestions and paper can be published. best regards